# NeurVPS: Neural Vanishing Point Scanning via Conic Convolution

**Yichao Zhou**[*]
UC Berkeley
zyc@berkeley.edu

**Haozhi Qi**
UC Berkeley
hqi@berkeley.edu

**Jingwei Huang**
Standford University
jingweih@stanford.edu

**Yi Ma**[†]
UC Berkeley
yima@eecs.berkeley.edu

## Abstract

We present a simple yet effective end-to-end trainable deep network with geometry-inspired convolutional operators for detecting vanishing points in images. Traditional convolutional neural networks rely on aggregating edge features and do not have mechanisms to directly exploit the geometric properties of vanishing points as the intersections of parallel lines. In this work, we identify a canonical conic space in which the neural network can effectively compute the global geometric information of vanishing points locally, and we propose a novel operator named *conic convolution* that can be implemented as regular convolutions in this space. This new operator explicitly enforces feature extractions and aggregations along the structural lines and yet has the same number of parameters as the regular 2D convolution. Our extensive experiments on both synthetic and real-world datasets show that the proposed operator significantly improves the performance of vanishing point detection over traditional methods. The code and dataset have been made publicly available at `https://github.com/zhou13/neurvps`.

## 1 Introduction

Vanishing point detection is a classic and important problem in 3D vision. Given the camera calibration, vanishing points give us the direction of 3D lines, and thus let us infer 3D information of the scene from a single 2D image. A robust and accurate vanishing point detection algorithm enables and enhances applications such as camera calibration [10], 3D reconstruction [18], photo forensics [35], object detection [19], wireframe parsing [48, 49], and autonomous driving [28].

Although there has been a lot of work on this seemingly basic vision problem, no solution seems to be quite satisfactory yet. Traditional methods (see [46, 27, 41] and references therein) usually first use edge/line detectors to extract straight lines and then cluster them into multiple groups. Many recent methods have proposed to improve the detection by training deep neural networks with labeled data. However, such neural networks often offer only a coarse estimate for the position of vanishing points [26] or horizontal lines [45]. The output is usually a component of a multi-stage system and used as an initialization to remove outliers from line clustering. Arguably the main reason for neural networks' poor precision in vanishing point detection (compared to line clustering-based methods) is likely because existing neural network architectures are not designed to represent or learn the special geometric properties of vanishing points and their relations to structural lines.

To address this issue, we propose a new convolutional neural network, called *Neural Vanishing Point Scanner (NeurVPS)*, that explicitly encodes and hence exploits the global geometric information of vanishing points and can be trained in an end-to-end manner to both robustly and accurately predict vanishing points. Our method samples a sufficient number of point candidates and the network then determines which of them are valid. A common criterion of a valid vanishing point is whether it lies on the intersection of a sufficient number of structural lines. Therefore, the role of our network

---

[*]We thank Yikai Li from SJTU and Jiajun Wu from MIT for their useful suggestions.

[†]This work is partially supported by the funding from Berkeley EECS Startup fund, Berkeley FHL Vive Center for Enhanced Reality, research grants from Sony Research, and Bytedance Research Lab (Silicon Valley).

is to measure the intensity of the signals of the structural lines passing through the candidate point. Although this notion is simple and clear, it is a challenging task for neural networks to learn such geometric concept since the relationship between the candidate point and structural lines not only depend on global line orientations but also their pixel locations. In this work, we identify a *canonical conic space* in which this relationship only depends on local line orientations. For each pixel, we define this space as a local coordinate system in which the x-axis is chosen to be the direction from the pixel to the candidate point, so the associated structural lines in this space are always horizontal.

We propose a *conic convolution operator*, which applies regular convolution for each pixel in this conic space. This is similar to apply regular convolutions on a rectified image where the related structural lines are transformed into horizontal lines. Therefore the network can determine how to use the signals based on local orientations. In addition, feature aggregation in this rectified image also becomes geometrically meaningful, since horizontal aggregation in the rectified image is identical to feature aggregation along the structural lines.

Based on the canonical space and the conic convolution operator, we are able to design the convolutional neural network that accurately predicts the vanishing points. We conduct extensive experiments and show the improvement on both synthetic and real-world datasets. With the ablation studies, we verify the importance of the proposed conic convolution operator.

## 2    Related Work

**Vanishing Point Detection.** Vanishing point detection is a fundamental and yet surprisingly challenging problem in computer vision. Since initially proposed by [3], researchers have been trying to tackle this problem from different perspectives. Early researches estimate vanishing points using sphere geometry [3, 30, 40], hierarchical Hough transformation [36], or the EM algorithms [46, 27]. Researches such as [43, 33, 4, 1] use the Manhattan world assumptions [12] to improve the accuracy and the reliability of the detection. [2] extends the mutual orthogonality assumption to a set of mutual orthogonal vanishing point assumption (Atlanta world [37]).

The dominant approach is line-based vanishing point detection algorithms, which are often divided into several stages. Firstly, a set of lines are detected [8, 42]. Then a line clustering algorithm [32] are used to propose several guesses of target vanishing point position based on geometric cues. The clustering methods include RANSAC [5], J-linkage [41], Hough transform [20], or EM [46, 27]. [50] uses contour detection and J-linkage in natural scenes but only one dominate vanishing point can be detected. Our method does not rely on existing line detectors, and it can automatically learn the line features in the conic space to predict any number of vanishing points from an image.

Recently, with the help of convolutional neural networks, the vision community has tried to tackle the problem from a data-driven and supervised learning approach. [9, 6, 47] formulate the vanishing point detection as a patch classification problem. They can only detect vanishing points within the image frame. Our method does not have such limitation. [45] detects vanishing points by first estimating horizontal vanishing line candidates and score them by the vanishing points they go through. They use an ImageNet pre-trained neural network that is fine-tuned on Google street images. [26] uses inverse gnomonic image and regresses the sphere image representation of vanishing point. Both work rely on traditional line detection algorithms while our method learns it implicitly in the conic space.

**Structured Convolution Operators.** Recently more and more operators are proposed to model spatial and geometric properties in images. For instance the wavelets based scattering networks (ScatNet) [7, 39] are introduced to ensure certain transform (say translational) invariance of the network. [22] first explores geometric deformation with modern neural networks. [14, 23] modify the parameterization of the global deformable transformation into local convolution operators to improve the performance on image classification, object detection, and semantic segmentation. More recently, structured and free-form filters are composed [38]. While these methods allow the network to learn about the space where the convolution operates on, we here explicitly define the space from first principle and exploit its geometric information. Our method is similar to [22] in the sense that we both want to rectify input to a canonical space. The difference is that they learn a global rectification transformation while our transformation is local. Different from [14, 23], our convolutional kernel shape is not learned but designed according to the desired geometric property.

Guided design of convolution kernels in canonical space is well practiced for irregular data. For spherical images, [11] designs operators for rotation-invariant features, while [24] operates in the

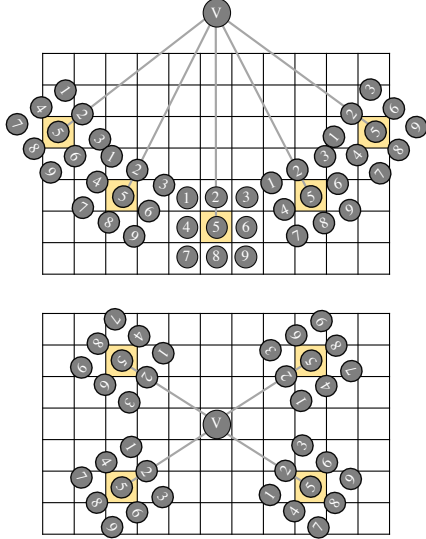

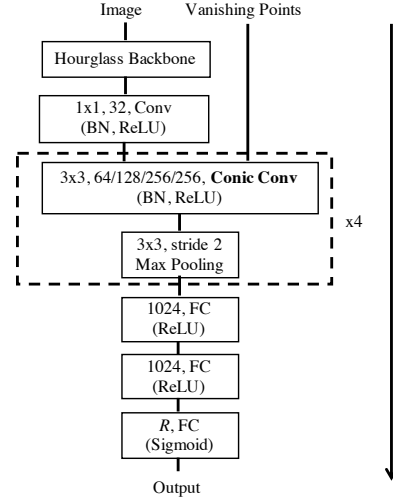

Figure 1: Illustration of sampled locations of $3 \times 3$ conic convolutions. The bright yellow region is the output pixel and $\mathbf{v}$ stands for the vanishing point. Upper and lower figures illustrate the cases when the vanishing point is outside and inside the image, respectively.

Figure 2: Illustration of the overall network structure. The number of each convolutional block is the kernel size and output dimension respectively. The number of fully connected layer block is the output dimension. The kernel size of Max Pooling layer is 3 and stride is 2. Batch normalization and ReLU activation are appended after each conv/fc layer except the last one use sigmoid as activation.

space defined by longitude and latitude, which is more meaningful for climate data. In 3D vision, geodesic CNN [31] adopts mesh convolution with the spherical coordinate, while TextureNet [21] operates in a canonical space defined by globally smoothed principal directions. Although we are dealing with regular images, we observe a strong correlation between the vanishing point and the conic space, where the conic operator is more effective than regular 2D convolution.

## 3 Methods

### 3.1 Overview

Figure 2 illustrates the overall structure of our NeurVPS network. Taken an image and a vanishing point as input, our network predicts the probability of a candidate being near a ground-truth vanishing point. Our network has two parts: a backbone feature extraction network and a conic convolution sub-network. The backbone is a conventional CNN that extracts semantic features from images. We use a single-stack hourglass network [34] for its ability to possess a large receptive field while maintaining fine spatial details. The conic convolutional network (Section 3.4) takes feature maps from the backbone as input and determines the existence of vanishing points around candidate positions (as a classification problem). The conic convolution operators (Section 3.3) exploit the geometric priors of vanishing points, and thus allow our algorithm to achieve superior performance without resorting to line detectors. Our system is end-to-end trainable.

Due to the classification nature of our model, we need to sample enough number of candidate points during inference. It is computationally infeasible to directly sample sufficiently dense candidates. Therefore, we use a coarse-to-fine approach (Section 3.5). We first sample $N_{\mathbf{d}}$ points on the unit sphere and calculate their likelihoods of being the line direction (Section 3.2) of a vanishing point using the trained neural network classifier. We then pick the top $K$ candidates and sample another $N_{\mathbf{d}}$ points around each of their neighbours. This step is repeated until we reach the desired resolution.

### 3.2 Basic Geometry and Representations of Vanishing Points

The position of a vanishing point encodes the line 3D direction. For a 3D ray described by $\mathbf{o} + \lambda \mathbf{d}$ in which $\mathbf{o}$ is its origin and $\mathbf{d}$ is its unit direction vector, its 2D projection on the image is

$$z \begin{bmatrix} p_x \\ p_y \\ 1 \end{bmatrix} = \underbrace{\begin{bmatrix} f & 0 & c_x \\ & f & c_y \\ & & 1 \end{bmatrix}}_{\mathbf{K}} \cdot (\mathbf{o} + \lambda \mathbf{d}), \tag{1}$$

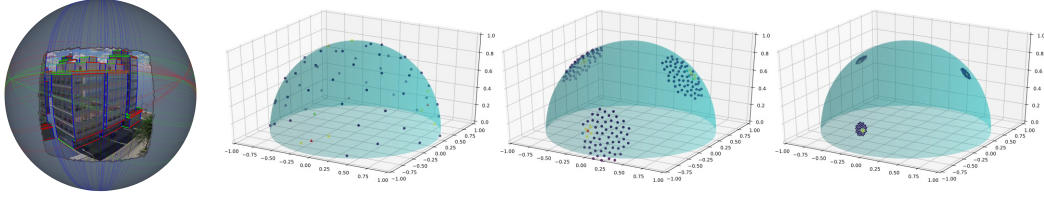

Figure 3: Illustration of vanishing points' Gaussian sphere representation of an image from the SU3 wireframe dataset [49] and our multi-resolution sampling procedure in the coarse-to-fine inference. In the right three figures, the red triangles represent the ground truth vanishing points and the dots represent the sampled locations.

where $p_x$ and $p_y$ are the coordinates in the image space, $z$ is the depth in the camera space, $\mathbf{K}$ is the calibration matrix, $f$ is the focal length, and $[c_x, c_y]^T \in \mathbb{R}^2$ is the optical center of the camera. The vanishing point is the point with $\lambda \to \infty$, whose image coordinate is $\mathbf{v} = [v_x, v_y]^T :=$ $\lim_{\lambda \to \infty}[p_x, p_y]^T \in \mathbb{R}^2$. We can then derive the 3D direction of a line in term of its vanishing point:

$$\mathbf{d} = [v_x - c_x \quad v_y - c_y \quad f]^T \in \mathbb{R}^3. \tag{2}$$

In the literature, a normalized line direction vector $\mathbf{d}$ is also called the *Gaussian sphere representation* [3] of the vanishing point $\mathbf{v}$. The usage of $\mathbf{d}$ instead of $\mathbf{v}$ avoids the degenerated cases when $\mathbf{d}$ is parallel to the image plane. It also gives a natural metric that defines the distance between two vanishing points, the angle between their normalized line direction vectors: $\arccos|\mathbf{d}_i^T\mathbf{d}_j|$ for two unit line directions $\mathbf{d}_i, \mathbf{d}_j \in \mathbb{S}^2$. Finally, sampling vanishing points with the Gaussian sphere representation is easy, as it is equivalent to sampling on a unit sphere, while it remains ambiguous how to sample vanishing points directly in the image plane.

### 3.3 Conic Convolution Operators in Conic Space

In order for the network to effectively learn vanishing point related line features, we want to apply convolutions in the space where related lines can be determined locally. We define the *conic space* for each pixel in the image domain as a rotated regular local coordinate system where the x-axis is the direction from the pixel to the vanishing point. In this space, related lines can be identified locally by checking whether its orientation is horizontal. Accordingly, we propose a new convolution operator, named *conic convolution*, which applies the regular convolution in this conic space. This operator effectively encodes global geometric cues for classifying whether a candidate point (Section 3.6) is a valid vanishing point. Figure 1 illustrates how this operator works.

A $3 \times 3$ conic convolution takes the input feature map $\mathbf{x}$ and the coordinate of convolution center $\mathbf{v}$ (the position candidates of vanishing points) and outputs the feature map $\mathbf{y}$ with the same resolution. The output feature map $\mathbf{y}$ can be computed with

$$\mathbf{y}(\mathbf{p}) = \sum_{\delta x=-1}^{1} \sum_{\delta y=-1}^{1} \mathbf{w}(\delta x, \delta y) \cdot \mathbf{x}(\mathbf{p} + \delta x \cdot \mathbf{t} + \delta y \cdot \mathbf{R}_{\frac{\pi}{2}}\mathbf{t}), \text{ where } \mathbf{t} := \frac{\mathbf{v} - \mathbf{p}}{\|\mathbf{v} - \mathbf{p}\|_2} \in \mathbb{R}^2. \tag{3}$$

Here $\mathbf{p} \in \mathbb{R}^2$ is the coordinates of the output pixel, $\mathbf{w}$ is a $3 \times 3$ trainable convolution filter, $\mathbf{R}_{\frac{\pi}{2}} \in \mathbb{R}^{2 \times 2}$ is the rotational matrix that rotates a 2D vector by $90°$ counterclockwise, and $\mathbf{t}$ is the normalized direction vector that points from the output pixel $\mathbf{p}$ to the convolution center $\mathbf{v}$. We use bilinear interpolation to access values of $\mathbf{x}$ at non-integer coordinates.

Intuitively, conic convolution makes edge detection easier and more accurate. An ordinary convolution may need hundreds of filters to recognize edge with different orientations, while conic convolution requires much less filters to recognize edges aligning with the candidate vanishing point because filters are firstly rotated towards the vanishing point. The strong/weak response (depends on the candidate is positive/negative) will then be aggregated by subsequent fully-connected layers.

### 3.4 Conic Convolutional Network

The conic convolutional network is a classifier that takes the image feature map $\mathbf{x}$ and a candidate vanishing point position $\hat{\mathbf{v}}$ as input. For each angle threshold $\gamma \in \Gamma$, the network predicts whether there exists a real vanishing point $\mathbf{v}$ in the image so that the angle between the 3D line directions $\mathbf{v}$ and $\hat{\mathbf{v}}$ is less than the threshold $\gamma$. The choice of $\Gamma$ will be discussed in Section 3.5.

Figure 2 shows the structure diagram of the proposed conic convolutional network. We first reduce the dimension for the feature map from the backbone to save the GPU memory footprint with an $1 \times 1$ convolution layer. Then 4 consecutive conic convolution (with ReLU activation) and max-pooling layers are applied to capture the geometric information at different spatial resolutions. The channel dimension is increased by a factor of two in each layer to compensate the reduced spatial resolution. After that, we flatten the feature map and use two fully connected layers to aggregate the features. Finally, a sigmoid classifier with binary cross entropy loss is applied on top of the feature to discriminate positive and negative samples with respect to different thresholds from $\Gamma$.

## 3.5 Coarse-to-fine Inference

With the backbone and the conic convolutional network, we can compute the probability of vanishing point over the hemisphere of the unit line direction vector $\hat{\mathbf{d}} \in \mathbb{S}^2$, as shown in Figure 3. We utilize a multi-resolution strategy to quickly pinpoint the location of the vanishing points. We use $R$ rounds to search for the vanishing points. In the $r$-th round, we uniformly sample $N_{\mathbf{d}}$ line direction vectors on the surface of the unit spherical cap with direction $\mathbf{n}_r$ and polar angle $\gamma_r$ using the Fibonacci lattice [17]. Mathematically, the $n$-th sampled line direction vector can be written as

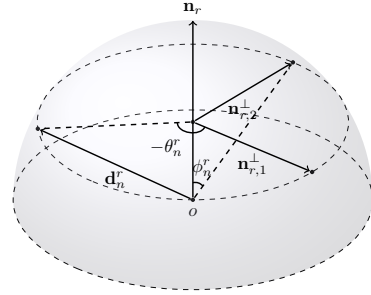

$$\mathbf{d}_n^r = \cos \phi_n^r \mathbf{n}_r + \sin \phi_n^r (\cos \theta_n^r \mathbf{n}_{r,1}^\perp + \sin \theta_n^r \mathbf{n}_{r,2}^\perp),$$
$$\phi_n^r := \arccos \left( 1 + (\cos \alpha_r - 1) * n / N_{\mathbf{d}} \right),$$
$$\theta_n^r := (1 + \sqrt{5})\pi n,$$

Figure 4: Illustration of the variables used in uniform spherical cap sampling.

in which $\mathbf{n}_{r,1}^\perp$ and $\mathbf{n}_{r,2}^\perp$ are two arbitrary orthogonal unit vectors that are perpendicular to $\mathbf{n}_r$, as shown in Figure 4. We initialize $\mathbf{n}_1 \leftarrow (0,0,1)$ and $\gamma_1 \leftarrow \pi$. For the round $r+1$, we set the threashold $\gamma_{r+1} \leftarrow \rho \max_{\mathbf{w} \in \mathbb{S}^2} \min_n \arccos |\langle \mathbf{w}, \mathbf{d}_n^r \rangle|$ and $\mathbf{n}_{r+1}$ to the $\mathbf{d}_n^r$ whose vanishing point obtains the best score from the conic convolutional network classifier with angle threshold $\gamma = \gamma_{r+1}$. Here, $\rho$ is a hyperparameter controlling the distance between two nearby spherical caps. Therefore, we set the threshold set $\Gamma$ in Section 3.3 to be $\{\gamma_{r+1} \mid r \in \{1, 2, \ldots, R\}\}$ accordingly.

The above process detects a single dominant vanishing point in a given image. To search for more than one vanishing point, one can modify the first round to find the best $K$ line directions and use the same process for each line direction in the remaining rounds.

## 3.6 Vanishing Point Sampling for Training

During training, we need to generate positive samples and negative samples. For each ground-truth vanishing point with line direction $\mathbf{d}$ and threshold $\gamma$, we sample $N^+$ positive vanishing points and $N^-$ negative vanishing points. The positive vanishing points are uniformly sampled from $\mathcal{S}^+ = \{\mathbf{w} \mid \mathbf{w} \in \mathbb{S}^2 : \arccos |\langle \mathbf{w}, \mathbf{d} \rangle| < \gamma\}$ and the negative vanishing points are uniformly sampled from $\mathcal{S}^- = \{\mathbf{w} \mid \mathbf{w} \in \mathbb{S}^2 : \gamma < \arccos |\langle \mathbf{w}, \mathbf{d} \rangle| < 2\gamma\}$. In addition, we sample $N^*$ random vanishing points for each image to reduce the sampling bias. The line directions of those vanishing points are uniformly sampled from the unit hemisphere.

# 4 Experiments

## 4.1 Datasets and Metrics

We conduct experiments on both synthetic [49] and real-world [50, 13] datasets.

**Natural Scene [50].** This dataset contains images of natural scenes from AVA and Flickr. The authors pick the images that contain only one dominating vanishing point and label their locations. There are 2,275 images in the dataset. We divide them into 2,000 training images and 275 test images randomly. Because this dataset does not contain the camera calibration information, we set the focal length to the half of the sensor width for vanishing point sampling and evaluation. Such focal length simulates the wide-angle lens used in landscape photography.

**ScanNet [13].** ScanNet is a 3D indoor environment dataset with reconstructed meshes and RGB images captured by mobile devices. For each scene, we find the three orthogonal principal directions for each scene which align with most of the surface normals and use them to compute the vanishing

points for each RGB image. We split the dataset as suggested by ScanNet v2 tasks, and train the network to predict the three vanishing points given the RGB image. There are 266,844 training images. We randomly sample 500 images from the validation set as our test set.

**SU3 Wireframe [49].** The "ground-truth" vanishing point positions in real world datasets are often inaccurate. To systematically evaluate the performance of our algorithm, we test our method on the recent synthetic SceneCity Urban 3D (SU3) wireframe dataset [49]. This photo-realistic dataset is created with a procedural building generator, in which the vanishing points are directly computed from the CAD models of the buildings. It contains 22,500 training images and 500 validation images.

**Evaluation Metrics.** Previous methods usually use horizon detection accuracy [2, 29, 45] or pixel consistency [50] to evaluate their method. These metrics are indirect for this task. To better understand the performance of our algorithm, we propose a new metric, called *angle accuracy* (AA). For each vanishing point from the predictions, we calculate the angle between the ground-truth and the predicted one. Then we count the percentage of predictions whose angle difference is within a pre-defined threshold. By varying different thresholds, we can plot the *angle accuracy curves*. $AA^\theta$ is defined as the area under the curve between $[0, \theta]$ divided by $\theta$. In our experiments, the upper bound $\theta$ is set to be $0.2°$, $0.5°$, and $1.0°$ on the synthetic dataset and $1°$, $2°$, and $10°$ on the real world dataset. Two angle accuracy curves (coarse and fine level) are plotted for each dataset. Our metric is able to show the algorithm performance under different precision requirements. For a fair comparison, we also report the performance in pixel consistency as in the dataset paper [50].

## 4.2 Implementation Detail

We implement the conic convolution operator in PyTorch by modifying the "im2col + GEMM" function according to Equation (3), similar to the method used in [14]. Input images are resized to $512 \times 512$. During training, the Adam optimizer [25] is used. Learning rate and weight decay are set to be $4 \times 10^{-4}$ and $1 \times 10^{-5}$, respectively. All experiments are conducted on two NVIDIA RTX 2080Ti GPUs, with each GPU holding 6 mini-batches. For synthetic data [49], we train 30 epochs and reduce the learning rate by 10 at the 24-th epoch. We use $\rho = 1.2$, $N^+ = N^- = 1$ and $N^* = 3$. For the Natural Scene dataset, we train the model for 100 epochs and decay the learning rate at 60-th epoch. For ScanNet [13], we train the model for 3 epochs. We augment the data with horizontal flip. We set $N_\mathbf{d} = 64$ and use $R_{\text{SU3}} = 5$, $R_{\text{NS}} = 4$, and $R_{\text{SN}} = 3$ in the coarse-to-fine inference for the SU3 dataset, the Natural Scene dataset, and the ScanNet dataset, respectively. During inference, the results from the backbone network can be shared so only the conic convolution layers need to be forwarded multiple times. Using the Nature Scene dataset as an example, we conduct 4 rounds of coarse-to-fine inference, in each of which we sample 64 vanishing points. So we forward the conic convolution part 256 times for each image during testing. The evaluation speed is about 1.5 vanishing points per second on a single GPU.

## 4.3 Ablation Studies on the Synthetic Dataset

**Comparison with Baseline Methods.** We compare our method with both traditional line detection based methods and neural network based methods. The sample images and results can be found in Figure 3 and supplementary materials. For line-based algorithms, the LSD with J-linkage clustering [42, 41, 16] probably is the most widely used method for vanishing point detection. Note that LSD is a strong competitor on the synthetic SU3 dataset as the images contain many sharp edges and long lines.

| | $AA^{0.2°}$ | $AA^{0.5°}$ | $AA^{1.0°}$ | mean | median |
|---|---|---|---|---|---|
| LSD [16] | 27.9 | 47.9 | 61.5 | 3.89° | 0.21° |
| REG | 2.2 | 6.5 | 15.0 | 2.07° | 1.48° |
| CLS | 2.2 | 9.1 | 23.7 | 1.77° | 0.99° |
| Conic×2 | 10.5 | 28.9 | 50.3 | 0.78° | 0.43° |
| Conic×4 | 47.5 | **74.2** | **86.3** | 0.15° | 0.09° |
| Conic×6 | **49.1** | 74.0 | 86.2 | **0.14°** | **0.09°** |

Table 1: Ablation study of our method. "REG" denotes the baseline that directly regress line direction in the camera space. "CLS" denotes the baseline that do vanishing point classification using image feature and its coordinate. Conic×$K$ denotes our methods with varying number of conic convolution layers.

We aim to compare pure neural network methods that only rely on raw pixels. Existing methods such as [9, 15, 6] can only detect vanishing points inside images. [45, 26] rely on an external line map as initial inputs. To the best of our knowledge, there is no existing pure neural network methods that are general enough to handle our case. Therefore, we propose two intuitive baselines. The first baseline, called REG, is a neural network that direct regresses value of $\mathbf{d}$ using chamfer-$\ell^2$ loss, similar to [49]. We change all the conic convolutions to traditional 2D convolutions to make the numbers of parameters be the same.

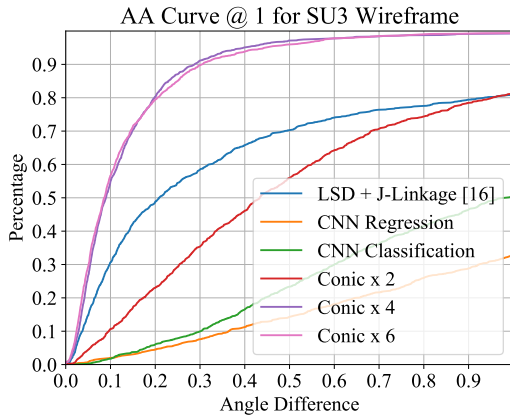
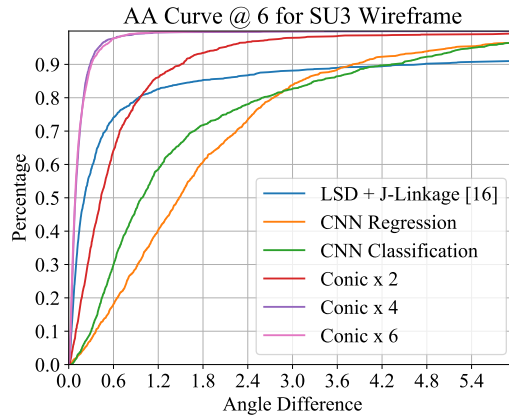

(a) Angle difference ranges from $0°$ to $1°$.

(b) Angle difference ranges from $0°$ to $6°$.

Figure 5: Angle accuracy curves for different methods on the SU3 wireframe dataset [49].

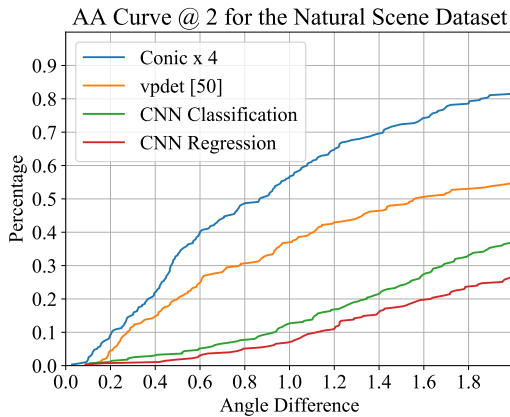
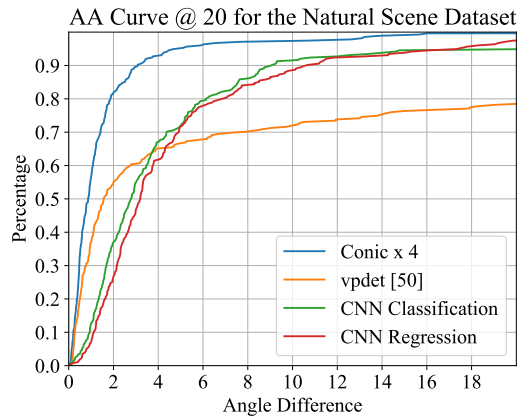

(a) Angle difference ranges from $0°$ to $2°$.

(b) Angle difference ranges from $0°$ to $20°$.

Figure 6: Angle accuracy curves for different methods on the Natural Scene dataset [50].

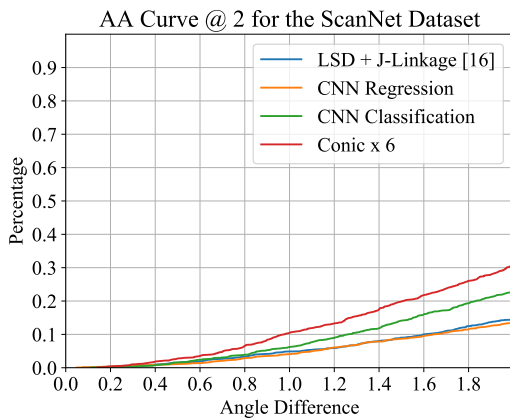
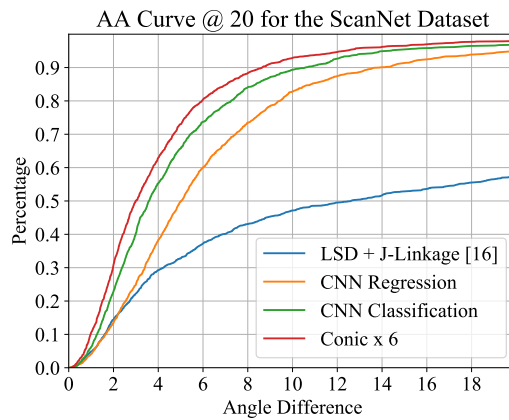

(a) Angle difference ranges from $0°$ to $2°$.

(b) Angle difference ranges from $0°$ to $20°$.

Figure 7: Angle accuracy curves for different methods on the ScanNet dataset [13].

The second baseline, called CLS, uses our fine-to-coarse classification approach. We change all the conic convolutions to their traditional counterparts, and concatenate **d** to the feature map right before feeding it to the NeurVPS head to make the neural network aware of the position of vanishing points.

The results are shown in Table 1 and Figure 5. By utilizing the geometric priors and large-scale training data, our method significantly outperforms other baselines across all the metrics. We note that, compared to LSD, neural network baselines perform better in terms of mean error but much worse for AA. This is because line-based methods are generally more accurate, while data-driven approaches are more unlikely to produce outliers. This phenomenon is also observed in Figure 5b, where neural network baselines achieve higher percentage when the angle error is larger than $4.5°$.

**Effect of Conic Convolution.** We now examine the effect of different numbers of conic convolution layers. We test with $2/4/6$ conic convolution layers, denoted as Conic$\times2/4/6$, respectively. For Conic$\times2$, we only keep the last two conic convolutions and replace others as their plain counterparts. For Conic$\times6$, we add two more conic convolution layers at the finest level, without max pooling appended. The results are shown in Table 1 and Figure 5. We observe that the performance keeps increasing when adding more conic convolutions. We hypothesize that this is because stacking multiple conic convolutions enables our model to capture higher order edge information and thus significantly increase the performance. The performance improvement saturates at Conic$\times6$.

## 4.4 NeurVPS on the Real World Datasets

**Natural Scene [50]** We validate our method on real world datasets to test its effectiveness and generalizability. The results of angle accuracy on the Natural Scene dataset [50] are shown in Table 2 and Figure 6. We also compare the performance in the *consistency measure*, a metric used by the baseline method (a contour-based clustering algorithm, labeled as vpdet) in the dataset paper [50] in Figure 8. Our method outperforms this strong baseline algorithm vpdet by a fairly large margin in term of all metrics. Our experiment also shows that the naive CNN baselines under-perform vpdet until the angle tolerance is around $4°$. This is consistent with the results from [50], in which vpdet is better than the previous deep learning method [45] in the region that requires high precision. Such phenomena indicates that our geometry-aware network is able to accurately locate vanishing points in images, while naive CNNs can only roughly determine vanishing points' position.

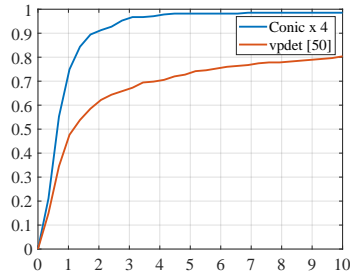

Figure 8: Consistency measure on the Nature Scene dataset [50].

**ScanNet [13]** The results on the ScanNet dataset [13] are shown in Table 3 and Figure 7. For baseline of traditional methods, we only compare our method with LSD + J-linkage because other methods such as [50] are not directly applicable when there are three vanishing points in a scene. Our results reduced the mean and median error by 6 and 4 times, respectively. The angle accuracy also improves by a large margin. The ScanNet [13] is a large dataset, so both CLS and REG works reasonable good. However, because the traditional convolution cannot fully exploit the geometry structure of vanishing points, the performance of those baseline algorithms is worse than the performance of our conic convolutional neural network. It is also worth mentioning that errors of ground truth

|  | $AA^{1°}$ | $AA^{2°}$ | $AA^{10°}$ | mean | median |
|---|---|---|---|---|---|
| REG | 2.4 | 9.9 | 58.8 | $5.09°$ | $3.20°$ |
| CLS | 4.4 | 14.5 | 62.4 | $5.80°$ | $2.79°$ |
| vpdet [50] | 18.5 | 33.0 | 60.0 | $12.6°$ | $1.56°$ |
| **Ours** | **29.1** | **50.3** | **85.5** | **$1.83°$** | **$0.87°$** |

Table 2: Performance of algorithms on the Natural Scene dataset [50]. vpdet is the method from the dataset paper.

|  | $AA^{1°}$ | $AA^{2°}$ | $AA^{10°}$ | mean | median |
|---|---|---|---|---|---|
| LSD [16] | 1.7 | 5.4 | 24.8 | $12.6°$ | $11.8°$ |
| REG | 1.5 | 5.1 | 45.1 | $6.9°$ | $5.0°$ |
| CLS | 2.0 | 8.1 | 55.9 | $5.3°$ | $3.6°$ |
| **Ours** | **3.4** | **11.5** | **61.7** | **$4.5°$** | **$3.0°$** |

Table 3: Performance of algorithms on ScanNet [13].

vanishing points of the ScanNet dataset are quite large due to the inaccurate 3D reconstruction and budget capture devices, which probably is the reason why the performance gap between conic convolutional networks and traditional 2D convolutional networks is not so significant.

One drawback of our data-driven method is the need of large amount of training data. We do not evaluate our method on datasets such as YUD [15], ECD [2], and HLW [44] because there is no suitable public dataset for training. In the future, we will study how to exploit geometric information under unsupervised or semi-supervised settings hence to alleviate the data scarcity problem.

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
