[Supplementary Material]

# A Supplementary Materials

Figure 10 and Figure 9 show the visual quality of the ground truth vanishing points and our predicted vanishing points on **random sampled** testing images in the Natural Scene dataset [50] and the SU3 wireframe dataset [49]. We display the images and three vanishing points of the SU3 wireframe dataset on Gaussian spheres because most of them are on the outside of the images, and we display the single dominating vanishing points in the natural scene dataset directly on the image.

Figure 9: Random sampled results from the natural scene dataset [50]. The red dots represent the ground truth vanishing points and the blue dots represent our predicted vanishing points.

Figure 10: Random sampled results from the synthetic SU3 wireframe dataset [49]. The lines on the sphere shows the ground truth lines and the colored dots shows the predicted vanishing points.