[Reviews · NeurIPS 2019]

Reviewer 1



Summary: This paper addresses the problem of estimating vanishing points from an RGB image. As opposed to traditional methods that rely on line detection and geometry, this paper proposes a data-driven method termed as Neural Vanishing Point Scanning (NeurVPS) to detect the location of the vanishing points, with the assumption that a camera calibration is given, on the 3D unit sphere (or homogeneous coordinate) w.r.t the image frame of reference. Specifically, the authors employ a neural network that takes as input an RGB image and a candidate vanishing point?s location on the 3D sphere. This network first extracts the embedding features from the RGB image using an hourglass architecture, then determines if there exists a true vanishing point that is close to the candidate input within a predefined angle distance. There are two challenges that this approach has to overcome: (i) To narrow down to the true vanishing point?s location, it needs to sample a lot of candidates on the 3D sphere which easily becomes an exhaustive search; and (ii) The accuracy of the standard convolution operator is not sufficient for detecting related lines? embedding features (i.e., lines that are parallel and hence intersect at a vanishing point at infinity) since it operates on the x- and y- spatial direction of the image frame while the vanishing points? locations are arbitrary. To address these challenges the paper makes the following contributions: - An adaptive vanishing points sampling method that: (i) In the inference phase, vanishing point candidates are iteratively sampled in the surface of the 3D unit sphere by the coarse-to-fine fashion. During which, the network determines which sample point(s) are closest to the true vanishing point location. Next, a finer sampling strategy is employed to extract more candidates that are near those determined from the previous step. The iterative process continues until it converges to the true vanishing point?s location; (ii) In the training phase, it additionally samples the negative and uniform batch to prevent the network from bias. - A conic convolution operator that is used to extract ?line features? guided by the candidate vanishing direction, in order to extract all the features in a canonical space (conic space) where all parallel lines that share the same vanishing point must follow the x-direction in the image frame of reference. This operator is then employed into the conic convolution network as a classification to determine if there is a true vanishing point nearby the candidate input. - An extensive experimental results on both synthetic and real datasets, comparing their method against the traditional approach with line detection and model fitting (J-Linkage), as well as data-driven approach (e.g., direct regression on vanishing point location, the proposed network with standard convolution, and multiple parameters of the conic convolution with multiple parameters) and their network that only employs standard convolution operator. Besides the mean and median angular error, they employed a histogram based statistics to count the percentage of the predicted vanishing points that have angular error less than 0.2, 0.5 and 1.0 degree on synthetic data and 1.0, 2.0 and 10.0 degree on the real dataset (similar to the metrics used in Surface Normal estimation literature [1]-[5]). Tables 1, 2, 3 and figures 5, 6, 7 show that their method outperforms other baseline on both synthetic and real dataset. As an example, in Table 1, the mean angular error using their method in the synthetic dataset can achieve sub-degree accuracy while the other methods cannot. Pros: - The proposed method is simple and effective, the experimental results section demonstrates that it outperforms the baselines (traditional approach and other data-driven methods) as indicated previously. - While the proposed classification network has an advantage over regression one since it can employ the conic convolution; however, to get high accuracy, the naive method suffers from sampling too many points. As a technical contribution, the coarse-to-fine sampling method improves the efficiency of the inference phase, which is critical for their proposed classification network. - The idea of conic convolution is similar to warping an image / embedding feature then employing the standard convolution (as indicated in the paper); but it is better defined for one vanishing point (to warp an image using homography, it requires at least 2 linear independent vanishing points). - This method also overcomes the main limitation of the traditional approach, where quality of vanishing point estimate highly depends on the quality of the line segment detection. - The paper is well-written in general. Cons: - While outperforming the standard convolution, the efficiency of the conic convolution is questionable. The author should provide more detail implementation of this operator using the available machine learning packages (PyTorch, Tensorflow, Cuda, etc.) and discuss insight for how to improve their current implementation (as stated in line 229-230 ?...our current implementation is far from optimized for efficiency) since this is the main contribution of the paper. - Another source of inefficiency stems from the oversampling, which has little to do with the classification network itself. A guided sampling based on traditional methods of LSD and J-Linkage, in my opinion, should reduce tremendously the inference time. - The authors claim (in line 275) that: "there is no existing method reported their performance on this (ScanNet) dataset". It might be true that no work has directly estimated 3 dominant vanishing directions, but there are a lot of work [1-4] (while the first 2 can generalized to train on ScanNet, the last 2 provides results on ScanNet) that estimate the surface normal from an rgb image. Clustering this estimated normal into 3 dominating directions, is a potential way to compare against the proposed method. This is exactly the same as one that the author proposed to obtain ground truth vanishing points for ScanNet. [1] Eigen, David, and Rob Fergus. "Predicting depth, surface normals and semantic labels with a common multi-scale convolutional architecture." In Proceedings of the IEEE international conference on computer vision, pp. 2650-2658. 2015. [2] Qi, Xiaojuan, Renjie Liao, Zhengzhe Liu, Raquel Urtasun, and Jiaya Jia. "Geonet: Geometric neural network for joint depth and surface normal estimation." In Proceedings of the IEEE Conference on Computer Vision and Pattern Recognition, pp. 283-291. 2018. [3] Ren, Zhongzheng, and Yong Jae Lee. "Cross-domain self-supervised multi-task feature learning using synthetic imagery." In Proceedings of the IEEE Conference on Computer Vision and Pattern Recognition, pp. 762-771. 2018. [4] Zhang, Zhenyu, Zhen Cui, Chunyan Xu, Yan Yan, Nicu Sebe, and Jian Yang. "Pattern-Affinitive Propagation across Depth, Surface Normal and Semantic Segmentation." In Proceedings of the IEEE Conference on Computer Vision and Pattern Recognition, pp. 4106-4115. 2019. Typos: - Line 226: ?the training date ?? -> ?the training data ?? - Line 237: ?We aimed to ?? -> ?We aim to ....? - Line 255: ?... We now exam ?? -> ?... We now examine ??

Reviewer 2



In this paper, the authors proposed a conic convolution operation for vanishing point scanning, my comments toward this paper are shown as follows: 1) The writing of this paper needs improvement. In section 1 and 2, the motivation and relationship with previous methods have not been introduced clearly. The English usage is also not good. 2) After sampling v in 3D space, how the direction vector calculated in the 2D space? 3) In the inference phase, the network need to forward many times, which will lead to a huge computational burden. Can the authors report more details about their coarse to fine algorithm, e.g. how many samples have been generated in the coarsely, how many samples have been selected to the next stage of processing, in the next stage of processing, how many samples will be generated around the seed samples? At the finest scale, in order to get pixel level accuracy, do you need to scan a large amount of pixels one by one? I think the speed of this algorithm should be very slow. 4) Although the authors argue their proposed metric is better than the previous ones, I think it will be better if the authors could also report conventional metrics on the competing datasets. 5) The authors tested the proposed algorithm on four datasets, but only compared with the proposed baseline methods, I think the authors should also report the baseline results provided by the dataset papers. Comments after rebuttal: The authors have addressed my questions on the efficiency and implementation details issues, but I still think the writing of this paper needs some improvement. Due to the above reasons, I raise my rating to 6.

Reviewer 3



Originality: The problem to solve in this paper is an existing topic, but the method is novel. They propose a novel framework as well as a novel operator to exact the geometric feature for vanishing points. Quality: The quality of this paper is good. - Although the loss for training the network is clearly described, I recommend the loss function should also be written in the paper. - The formula (3) is kind of confusing: e.g., delta x and delta y are not clearly defined. Clarity: The paper is clear. Significance: This work is relatively significant.

[Author Response · NeurIPS 2019]

## 1. Common Questions

**Coarse-to-Fine Inference:** During inference, the results from the backbone network can be shared so only the conic convolution layers need to be forwarded multiple times. Using the Nature Scene dataset as an example, we conduct 4 rounds of coarse-to-fine inference, in each of which we sample 64 vanishing points. So we forward the conic convolution part 256 times for each image during testing. We will clarify such implementation details in the revision.

**Improving Efficiency:** We note that the speed of academic implementation of LSD/J-Linkage is also 1 FPS, while Contour/J-Linkage takes more than two minutes per image. On the Nature Scene dataset, we have achieved 2.8 FPS with an RTX 2080 Ti by tweaking the block size of im2col and removing one round of inference. The median error increases from 1.10 to 1.11. We may further optimize the algorithm by decreasing the sample number but using more rounds in inference, and adding bottleneck architectures and group convolution to reduce the ops of $3 \times 3$ conic convolution.

## 2. Response to Reviewer #1

**Implementation:** Our conic convolution operator is implemented by modifying the "im2col + GEMM" function, which is used to implement ordinary convolution in Caffe, MxNet, and Tensorflow, etc. We change the sampling locations of im2col function according to Equation (3). A similar implementation is also used in [1].

**Geometry Insight:** Conic convolution makes edge detection easier and more accurate. An ordinary convolution may need hundreds of filters to recognize edge with different orientations, while conic convolution requires much less filters to recognize edges aligning with the candidate vanishing point because filters are firstly rotated towards the vanishing point. The strong/weak response (depends on the candidate is positive/negative) will then be aggregated by the subsequent fully-connected layer.

Figure I: More results on ScanNet.

**Guided Sampling Inference:** We intentionally keep our method simple so that it can be fully trained in an end-to-end fashion. Leveraging intermediate results from traditional methods to the inference process could possibly speed up the computation, but it may also inherit bias and failure modes of such methods. This is nevertheless a valuable direction to study in the future.

**Normal Clustering Baseline:** We have conducted the suggested experiment that clusters the network-estimated surface normals into 3 principle directions using the pre-trained model from FrameNet [2], a state-of-the-art normal prediction network on ScanNet. The result is shown in Figure I. The normal clustering method under-performs other neural network baselines because it is not trained in an end-to-end fashion.

## 3. Response to Reviewer #2

**Projecting Line Direction Vector to 2D:** Line direction vector $\mathbf{d} \in \mathbb{R}^3$ are defined in the 3D camera space (line 117) and its 2D projection $\mathbf{v}$ to the image is the vanishing point (line 121). We can compute $\mathbf{v}$ by plugging $\mathbf{d}$ into Equation (1) with $\lambda \to \infty$, and then $\mathbf{v} = (p_x, p_y)$.

**Metrics from Dataset Papers:** We show the curves of consistency measure in Figure II, generated by the code from the authors of the Nature Scene dataset.

**Baselines from Dataset Papers:** On the Natural Scene dataset, we did compare our methods with the one from the dataset paper (line 265). On the SU3 dataset, our baseline network is the same as the one from the dataset paper except for the number of stacks and head networks (for fair comparison). In addition, LSD/J-Linkage is the de facto standard baseline for vanishing point detection.

Figure II: Consistency measure on the Nature Scene dataset.

## 4. Response to Reviewer #3

**Clarification:** We will write down the loss function in the revision: $L(\hat{y}, y) = -(y \log(\hat{y}) + (1 - y) \log(1 - \hat{y}))$, where $y$ is the classification label and $\hat{y}$ is the network prediction. Besides, we will clarify that $\delta x$ and $\delta y$ in Equation (3) are defined under the summation symbol, representing the offset relative to the convolution center.

**Regarding Triplet Loss:** Triplet loss is typically used to learn a similarity metric among instances whereas our task is a binary classification problem. It is difficult to pinpoint position of a vanishing point based only on similarity scores.

**Regarding Metrics:** We did report the mean angle error in the "mean" columns in Tables 1-3 of the paper. However, we find that this metric is unfair to traditional methods because outliers would dominate such errors. For example, in Table 1, the mean error of LSD is much higher than the error of the neural network baseline, but in general the neural network baseline is more inaccurate, according to Figure 5(a).

[1] Jifeng Dai, Haozhi Qi, Yuwen Xiong, Yi Li, Guodong Zhang, Han Hu, and Yichen Wei. Deformable convolutional networks. In *ICCV*, 2017.

[2] Jingwei Huang, Yichao Zhou, Thomas Funkhouser, and Leonidas Guibas. FrameNet: Learning local canonical frames of 3D surfaces from a single RGB image. *arXiv preprint arXiv:1903.12305*, 2019.


[Meta-Review · NeurIPS 2019]

Three reviewers recommend acceptance. This is a good paper and should be accepted.